# Mendelian Randomization Analysis Reveals Causal Effects of Polyunsaturated Fatty Acids on Subtypes of Diabetic Retinopathy Risk

**DOI:** 10.3390/nu15194208

**Published:** 2023-09-29

**Authors:** Shaojie Ren, Chen Xue, Manhong Xu, Xiaorong Li

**Affiliations:** Tianjin Key Laboratory of Retinal Functions and Diseases, Tianjin Branch of National Clinical Research Center for Ocular Disease, Eye Institute and School of Optometry, Tianjin Medical University Eye Hospital, Tianjin 300384, China; reneeeooo3@gmail.com (S.R.); 15062262676@163.com (C.X.); hollyxuu@gmail.com (M.X.)

**Keywords:** diabetic retinopathy, fatty acids, omega-3, omega-6, Mendelian randomization

## Abstract

Polyunsaturated fatty acids (PUFAs) affect several physiological processes, including visual acuity, but their relationship with diabetic retinopathy (DR) remains elusive. The aim of this study was to determine whether PUFAs have a causal effect on DR. PUFAs- (total and omega-3 [FAw3] and omega-6 [FAw6] fatty acids and their ratio) and DR-associated single nucleotide polymorphisms derived from genome-wide association studies; sample sizes were 114,999 for fatty acids and 216,666 for any DR (ADR), background DR (BDR), severe non-proliferative DR (SNPDR), and proliferative DR (PDR). We hypothesized that the intra-body levels of PUFAs have an impact on DR and conducted a two-sample Mendelian randomization (MR) study to assess the causality. Pleiotropy, heterogeneity, and sensitivity analyses were performed to verify result reliability. High levels of PUFAs were found to be associated with reduced risk of both ADR and PDR. Moreover, FAw3 was associated with a decreased risk of PDR, whereas FAw6 demonstrated an association with lowered risks of both BDR and PDR. Our findings provide genetic evidence, for the first time, for a causal relationship between PUFAs and reduced DR risk. Consequently, our comprehensive MR analysis strongly urges further investigation into the precise functions and long-term effects of PUFAs, FAw3, and FAw6 on DR.

## 1. Introduction

Diabetic retinopathy (DR), categorized into non-proliferative and proliferative stages (NPDR and PDR, respectively), constitutes the principal ocular sequel of diabetes mellitus, impacting approximately 30–40% of individuals with diabetes [1,2]. Globally, over 132 million individuals are affected by diabetic retinopathy (DR), a significant driver of visual impairment and blindness, especially within the working-age adult demographic [1,3]. The worldwide occurrence and impact of DR are projected to experience a substantial escalation over the next few decades, increasing from around 103 million cases in 2020 to 130 million by 2030 and further to 161 million by 2045 [4]. This sudden rise in the prevalence of DR by more than 25% in just 10 years is likely to put additional strain on the already-overburdened healthcare systems and resources. The financial expenses related to DR and its complications are substantial [5].

Polyunsaturated fatty acids (PUFAs), encompassing both omega-3 fatty acids (FAw3) and omega-6 fatty acids (FAw6), are considered to have numerous beneficial effects due to their anti-inflammatory and anti-aging properties. Consequently, the supplementation of PUFAs in the diet may serve as an effective adjunct therapy, particularly in the context of various diseases that are rooted in aging, cellular dysfunction, and prolonged exposure to light, notably eye diseases [6]. As essential constituents of the retina, PUFAs play a pivotal role in ensuring the survival of photoreceptors and actively participating in visual signal transmission, thereby maintaining the dynamic equilibrium and promoting the overall health of the retina [7,8,9]. Recently, high levels of PUFAs, particularly FAw3, have been found to be associated with a reduced risk of DR progression [10,11,12,13], whereas controversies persist regarding the effects of FAw6 [14,15]. Furthermore, recognizing that the pathological mechanisms underlying early-stage DR and advanced DR differ significantly is imperative [16]. A comprehensive understanding of these distinctions is crucial for devising tailored strategies for both prevention and treatment. However, it is worth noting that existing research on PUFAs and DR primarily focuses on a holistic perspective, lacking a refined classification based on different stages of the condition.

Serum lipid profiles are considerably altered by variables such as fasting and medication use, independent of the underlying disease, and conventional studies cannot completely rule out confounding bias or reverse causality [17,18]. Randomized controlled trials (RCTs) are typically regarded as the gold standard in clinical evidence [19], but to date, no RCTs have examined the effect of PUFAs on DR, possibly owing to difficulties in meeting the criteria of sample size and intervention duration. Mendelian randomization (MR) has become a popular and practical approach, employed mainly for epidemiological etiological inference in recent years, and can imitate RCTs without much difficulty [20,21]. It employs genetic variation as an instrumental variable (IV) to ascertain causal relationships between biomarkers and outcomes [22]. Utilizing the inherent randomization provided by genetic variation, this approach circumvents the issues associated with confounding factors and reverse causation. It serves as a suitable method to assess the connection between serum lipid levels and relevant diseases [23].

In this study, we hypothesized that the intracellular levels of PUFAs have an impact on different subtypes of DR. To prove the hypothesis, we estimated bidirectional causal effects between total PUFAs and any DR (ADR), and performed a 2-sample MR study to comprehensively investigate the potential causality of different types of PUFAs (total PUFAs, FAw3, FAw6, and FAw6/FAw3) on three DR phenotypes—background DR (BDR), severe NPDR (SNPDR), and PDR—to assess the associations between PUFAs and DR.

## 2. Materials and Methods

### 2.1. Study Design

In this study, we conducted a bidirectional MR analysis utilizing summary genetic associations from various genome-wide association studies (GWAS) for both total PUFAs and ADR. Next, we used 2-sample MR to examine the associations of different genetically determined PUFA traits (total PUFAs, FAw3, FAw6, and ratio of FAw6 to FAw3) with three DR phenotypes (BDR, SNPDR, and PDR). A schematic diagram outlining the process of the two-sample MR study investigating the relationship between PUFAs and DR is presented in Figure 1. Furthermore, we have adhered to the MR-STROBE guidelines in reporting our findings.

### 2.2. Data Source

#### 2.2.1. Exposure Data

The most recent and extensive GWAS summary statistics for PUFAs were obtained from the UK Biobank, a population-based cohort comprising around 500,000 individuals (approximately 5% of those invited). These statistics were extracted from the IEU database (https://gwas.mrcieu.ac.uk/, accessed on 1 May 2023). The GWAS for PUFAs contained 114,999 samples, all of which were from Europe, and patients ranged in age from 40 to 69 years [24]. This GWAS explains 4.8–7.9% of the variance in cyclic PUFAs.

Circulating concentrations of FAw3 and FAw6 were quantified using a targeted high-throughput nuclear magnetic resonance metabolomics platform (Nightingale Health Ltd., Helsinki, Finland; biomarker quantification version 2020) [25]. Pre-release data from a randomly selected subset of 126,846 non-fasting plasma samples collected at baseline or during the first repeat assessment were provided to early access analysts. Subsequently, 121,577 samples were retained for analysis after the removal of duplicates and observations that did not meet quality control criteria.

In this study, four PUFA traits were selected as exposure amounts: total PUFAs, circulating FAw3, FAw6, and the ratio of FAw6 to FAw3. The average levels of circulating fatty acids among participants in the UK Biobank were 0.53 mmol/L (SD: 0.22) for total FAw3 and 4.45 mmol/L (SD: 0.68) for FAw6. These concentrations corresponded to 4.4% and 38% of the total fatty acids, respectively.

#### 2.2.2. Outcome Data

The DR GWAS data were downloaded from the FinnGen research project (https://r5.finngen.fi/, accessed on 1 May 2023) [26]. Population isolates like Finland offer advantages in genetic research, concentrating damaging alleles on low-frequency variants due to historical bottlenecks. FinnGen, studying 500,000 Finns, has enriched data for rare diseases and late-onset conditions. The project included a GWAS on 1932 clinical endpoints, identifying genome-wide significant associations at 2491 loci, including 148 putative causal coding variants, with 62 having over two-fold enrichment in Finland and a low allele frequency (<10%) [27]. DR was identified using ICD-10 codes (H36.0 or E11.3). A total of 16,962,023 variants were analyzed from 218,792 subjects through SAIGE GWAS (https://github.com/weizhouumich/SAIGE, accessed on 1 May 2023). Four DR phenotypes were selected as exposures based on different levels of DR severity: ADR (ICD 10: H36.0*; 18,097 cases and 206,364 controls), BDR (ICD 10: H36.00; 2510 cases and 242,308 controls), SNPDR (ICD 10: H36.02; 568 cases and 242,308 controls), and PDR (ICD 10: H36.03; 10,860 cases and 242,308 controls).

For the analysis, the possible confounders of age, sex, genotyping batches, genetic correlation, diabetes course, hypertension and cardiovascular history, and average daily dosage of hypoglycemic agents after cohort entry were adjusted. Further details on FinnGen are described in https://finngen.gitbook.io/documentation/v/r7/, accessed on 1 May 2023. The FinnGen study is a Finnish national meta-analysis of GWAS from nine biobanks and has limited overlap with the UK Biobank GWAS from several centers in the UK. Therefore, we believe that the limited sample overlap between the PUFA and DR data leads to minimal risk of bias.

### 2.3. Selection of Genetic Instruments

We selected single-nucleotide polymorphisms (SNPs), representing instrumental variables (IVs), after screening, in accordance with the three core assumptions of MR: (i) the “correlation” hypothesis, in which IVs are closely related to exposure; (ii) the assumption of “independence”, in which IVs are not related to confounders; and (iii) the “exclusion limitation” hypothesis, in which IVs do not affect results by means other than exposure, namely, excluding horizontal pleiotropy [24]. A concise outline of the MR design is presented in Figure 1.

To fulfill the three fundamental assumptions of MR, we established the following criteria to screen SNPs used as IVs:(1)All SNPs underwent screening at the genome-wide significance threshold (*p* < 5 × 10^−8^).(2)Using the “ld_clump” R package, linkage disequilibrium between SNPs (R^2^ < 0.001 and <10,000 from the index variant) was identified to ensure their independence [28].(3)We aligned effect alleles of outcome-related SNPs with those of exposure-related SNPs based on allelic letters and frequencies and removed palindromic SNP alleles [28].(4)We used the PhenoScanner database (https://www.phenoscanner.medschl.cam.ac.uk/, accessed on 1 May 2023) to verify whether the SNP loci were associated with other confounding factors [29].

### 2.4. MR Methods

Prior to conducting the MR analysis, we computed F statistics for these IVs to assess the presence of weak IV bias. In all instances, we observed that F > 10. Therefore, owing to the diminished impact of weak IV bias, the chosen SNPs were utilized in the MR study [30].

In our study, we applied three complementary approaches—inverse-variance-weighted (IVW), MR-Egger regression, and weighted median (WM)—to assess the causal effects of exposures on outcomes. The primary outcome was obtained using the IVW method. This technique involves assigning weights based on the inverse variance of each study, followed by calculating the weighted average effect size and summarizing the effect sizes from multiple independent studies. When all selected SNPs are valid IVs, this method yields the most accurate estimation results [31]. To enhance the robustness of the IVW estimates, we employed the WM and MR-Egger methods. WM is the median of the weighted empirical distribution function of individual SNP ratio estimates. Approximately 50% of the weights in the analysis come from ratio estimates smaller than or equal to the WM. This method provides a consistent effect estimate if more than 50% of the information comes from valid SNPs [32]. MR-Egger allows all genetic variants to have a pleiotropic effect but requires that the pleiotropic effects be independent of the variant–exposure association [33]. Although WM and MR-Egger methods generally offer more reliable estimates across a wider range of scenarios, these can be less efficient. Ultimately, the causal estimates were expressed as odds ratios (ORs) along with their corresponding 95% confidence intervals (CIs).

### 2.5. Statistical Analysis

The “TwoSampleMR” (https://github.com/mrcieu/TwoSampleMR, accessed on 1 May 2023) and MR Pleiotropy RESidual Sum and Outlier (MR-PRESSO, version 4.2.0) packages in R (version 3.6.1; The R project for statistical computing) were used for all analyses. For the bidirectional MR of total PUFAs and ADR, a *p*-value < 0.05 on both sides indicates significance. The Bonferroni correction, a conservative method, was used to examine the associations between exposure (i.e., total PUFAs, FAw3, FAw6, and FAw6/FAw3) and DR outcomes (BDR, SNPDR, and PDR). For region-level analyses, a Bonferroni-corrected *p*-value < 4.17 × 10^−3^ [0.05/(3 × 4)] on both sides is regarded to indicate significance, a *p*-value < 0.05 on both sides is regarded to indicate nominal significance, and a *p*-value between 4.17 × 10^−3^ and 0.05 is considered as suggestive evidence.

We corrected horizontal pleiotropy by detecting and removing outliers using the MR-PRESSO test and determined whether a substantial change in causality before and after removing outliers was present [34]. MR-Egger regression was then used to evaluate the possibility of average horizontal pleiotropy among the IVs, and funnel charts were used to visualize directional pleiotropy. If the MR-Egger intercept is not statistically significant (*p* > 0.05), the relative symmetry of funnel plots can be considered as an indicator of horizontal pleiotropy [33,35].

Additionally, we employed Cochran’s Q test (significance set at *p* > 0.05 to indicate no heterogeneity) to evaluate potential heterogeneity among SNPs in IVW estimates. To ensure the robustness and consistency of the findings, we conducted a sensitivity analysis using a “leave-one-out” approach. Horizontal pleiotropy is corrected by removing outliers, which involved performing MR again with the omission of one SNP at a time [36].

## 3. Results

### 3.1. IVs for PUFAs and DR

Our rigorous screening process was based on the independence and exclusivity hypotheses as well as on the harmonization and removal of palindromic SNPs with intermediate allele frequencies. We thus meticulously selected 52 SNPs for MR analysis to assess the associations between total PUFA and ADR. Similarly, to evaluate the relationship between ADR and total PUFAs, eight SNPs were selected as IVs. In this study, we also investigated the causal effects of different types of PUFAs (total PUFA, FAw3, FAw6, and FAw6/FAw3) on three specific DR phenotypes (BDR, SNPDR, and PDR) using MR analysis. In total, 34–57 SNPs were chosen for this analysis. The F-statistics of the instruments exceeded 10 for all selected SNPs, indicating satisfactory instrument strength. Detailed information regarding the instruments is provided in Appendix A.

### 3.2. Bidirectional Causal Effects between Total PUFAs and ADR

The MR results revealed a substantial association between the elevated genetically predicted total PUFA levels and a reduced risk of ADR (IVW: OR, 0.90; 95% CI, 0.83–0.97; *p* = 5.05 × 10^−3^; MR-Egger: OR, 0,86; 95% CI, 0.74–1.00; *p* = 0.05; WM: OR, 0,90; 95% CI, 0.81–1.00; *p* = 0.05). The scatter plot (Figure 2) shows that the causal effects among the IVW, MR-Egger regression, and WM methods were consistent. However, no association between ADR and total PUFAs was found using any of the methods (all *p* > 0.05). Additional details for the MR analysis are shown in Figure 3.

### 3.3. Causal Effects of Different Types of PUFAs on Three DR Phenotypes

The results of MR analysis showed that genetically-predicted total PUFAs were associated with decreased risk of BDR (IVW: OR, 0.79; 95% CI, 0.65–0.98; *p* = 2.90 × 10^−2^; MR-Egger: OR, 0.97; 95% CI, 0.65–1.45; *p* = 0.87; WM: OR, 0.75; 95% CI, 0.57–1.00; *p* = 0.05) and PDR (IVW: OR, 0.86; 95% CI, 0.78–0.95; *p* = 2.63 × 10^−3^; MR-Egger: OR, 0.86; 95% CI, 0.72–1.04; *p* = 0.13; WM: OR, 0.88; 95% CI, 0.77–1.01; *p* = 0.08). FAw3 was associated with decreased risk of PDR (IVW: OR, 0.83; 95% CI, 0.74–0.93; *p* = 1.50 × 10^−3^; MR-Egger: OR, 0.82; 95% CI, 0.67–1.00; *p* = 0.06; WM: OR, 0,81; 95% CI, 0.70–0.94; *p* = 0.02) and FAw6, with decreased risk of BDR (IVW: OR, 0.69; 95% CI, 0.56–0.86; *p* = 9.20 × 10^−4^; MR-Egger: OR, 0.82; 95% CI, 0.53–1.27; *p* = 0.37; WM: OR, 0,65; 95% CI, 0.47–0.90; *p* = 9.25 × 10^−3^) and PDR (IVW: OR, 0.82; 95% CI, 0.74–0.91; *p* = 3.51 × 10^−4^; MR-Egger: OR, 0.89; 95% CI, 0.72–1.10; *p* = 0.28; WM: OR, 0.82; 95% CI, 0.69–0.96; *p* = 0.1). The causal effects using IVW, MR-Egger regression, and WM methods were consistent, as shown in the scatter plot (Figure 2). However, no association between any of the three fatty acids and SNPDR was found using any of the methods (all *p* > 0.05). In addition, no association between the combined FAw6/FAw3 ratio and any of the DR phenotypes was found using any of the methods (all *p* > 0.05). Further details of the MR analysis are shown in Figure 3.

### 3.4. Sensitivity Analysis

To bolster the credibility of the aforementioned outcomes, our assessment through MR-Egger regression analysis yielded no evidence of directional pleiotropy or heterogeneity (all *p* > 0.05), as depicted in Figure 3. Additionally, funnel plots were used to visually represent directional pleiotropy, displayed a symmetrical distribution in Appendix A. Moreover, after outlier SNPs were deleted, the application of MR-PRESSO, along with the examination of leave-one-out and funnel plots (Appendix A), did not reveal any outliers.

## 4. Discussion

In this study, we used GWAS data to perform bidirectional MR analyses based on the genetic associations of overall PUFAs and DR and conducted a two-sample MR study to examine the causal relationship between genetically determined PUFA features (overall PUFAs, FAw3, FAw6, and FAw6/FAw3) and three DR phenotypes (BDR, SNPDR, and PDR).

Our main findings, along with the sensitivity analyses, demonstrated that a genetic predisposition to higher levels of overall PUFAs was associated with a reduced risk of DR. Reverse MR analysis did not provide any genetic evidence of ADR influencing PUFAs. The total levels of PUFAs were associated with a decreased risk of BDR and PDR. Regarding specific fatty acid levels, elevated FAw3 levels were found to be linked to decreased risk of PDR, whereas increased FAw6 levels were associated with reduced risk of both BDR and PDR.

PUFAs are indispensable fatty acids that the human body cannot synthesize and must be acquired through dietary intake, and are found abundantly in fatty fish, nuts, and seeds [37]. PUFAs can be classified into two main types: FAw3 and FAw6 [13,38]. In recent years, numerous observational studies have explored the role of PUFAs in DR development [39,40,41]. One prospective study by Sala-Vila et al. [39] provided initial evidence of a reduced risk of severe DR in middle-aged and older individuals with type 2 diabetes who consumed fish oil. Similarly, Knud Erik Alsbirk et al. [10] conducted an observational study in a coastal population that exhibited high fish oil intake, comprising 510 Norwegian individuals with diabetes, and they observed that none of the participants experienced a decline in best corrected visual acuity to 0.3 (0.48 logMAR) due to DR, suggesting a potential protective effect of PUFAs against the incidence and progression of DR microvascular complications. Animal studies have also provided convincing evidence of the protective effects of PUFAs on DR development [39,40]. However, a few studies have reported no association or even harmful effects of PUFAs on DR [42,43]. Currently, there are only a few long-term randomized studies that have investigated the relationship between PUFA intake and DR. This scarcity may be attributed to the challenges in controlling for confounding factors in lipid measurements, such as age, sex, body mass index, medication usage, fasting status, and dietary habits [41,42]. The MR approach, which we employed in this study, not only mimics RCTs but also offers a suitable method for evaluating the relationship between blood lipids and the disease of interest. MR avoids confounding factors and reverse causality without requiring interventions on humans or animals. It is commonly used in epidemiological causal inference investigations and serves as a genetic validation tool for observational studies [36,44]. Our findings align with the majority of research findings that suggest a protective effect of high levels of total PUFAs on DR.

FAw3 primarily comprise long-chain omega-3 fatty acids, including eicosapentaenoic acid (EPA) and docosahexaenoic acid (DHA) [45]. Accumulating evidence suggests that supplementation with FAw3 benefits patients with DR [40,41]. Nevertheless, no studies have investigated the relationship between FAw3 intake and different DR subtypes. Our MR study aimed to address this knowledge gap. The findings indicated a potential mitigating effect of FAw3 on PDR, whereas FAw3 had no protective causal relationship on BDR or SNPDR, consistent with prior research linking increased FAw3 intake to reduced risk of severe visual decline. Moreover, FAw3 intake lowers the risk of pathological retinal neovascularization and severe visual impairment [11,46,47,48]. Notably, lipotoxins, hemolysins, and protective proteins derived from FAw3 exhibit anti-angiogenic properties and hold clinical potential for preventing diabetic macular edema and retinopathy [49].

The estimates provided in this study also consider the long-term effects of FAw6, which remain an important question to be addressed. Evidence regarding the association between FAw6 levels and the risk of DR is limited and is the subject of ongoing debate. Linoleic acid (LA), an FAw6, possesses anti-inflammatory properties and can modulate inflammation through various pathways, including immune cell function, and inhibition of production of inflammatory mediators among other actions [50]. Arachidonic acid (AA), another FAw6, plays a significant role in inflammatory responses and can be metabolized by enzymes such as cyclooxyuigenase and lipoxygenase to generate various inflammatory mediators, including prostaglandins and leukotrienes [50]. In contrast to FAw3, FAw6 are considered to have pro-inflammatory properties, and thus, maintaining a lower FAw6/FAw3 ratio is crucial for achieving favorable outcomes in ocular pathology [51]. However, studies on healthy adults have demonstrated that increasing intake of AA or LA does not necessarily result in elevated concentrations of several inflammatory markers [52,53,54]. Epidemiological research has even suggested that AA and LA are associated with reduced inflammation [42,55]. Additionally, a study by Fu et al. [56] showed that a diet rich in FAw6 accelerated the maturation process of retinal neurons while inducing increased metabolism to maintain the energy balance of retinal neurons. A study conducted by Kai Wang et al. [57] revealed that LA plays a significant role in the negative correlation between FAw6 and susceptibility to age-related macular degeneration. The controversy surrounding FAw6 may arise from the differential roles of LA and AA in the eye.

Based on the results of our MR study, we suggest a consistent mitigating effect of FAw6, including LA and AA, on the risk of PDR. Of note, compared with FAw3, FAw6 also exhibited a mitigating effect on the risk of BDR, although neither type had an effect on SNPDR. This suggests an overall beneficial role for FAw6 in DR, although the specific mechanisms are yet to be studied. The interaction between FAw3 and FAw6, as well as their lipid mediators, in the context of inflammation is complex and not yet fully understood [55,58,59]. Lastly, we did not find any significant influence of the FAw6 to FAw3 ratio on any DR subtype.

Notably, dietary sources of PUFAs are limited in Western diets, which highlights the importance of dietary interventions or nutritional supplementation, especially in pregnant women, children, and older populations [60]. Therefore, implementing simple measures to enhance the production of endogenous anti-inflammatory molecules and adopting seemingly straightforward interventions throughout the entire course of DR in patients with long-term diabetes may be beneficial [37]. The establishment of a therapeutic approach for administering FAw3 to treat DR remains an important unanswered question. Therefore, further research is encouraged to determine the specific role and long-term effects of FAw3 and FAw6 in the treatment of DR.

### Limitations

This study has certain limitations. First, it is important to highlight that while the GWAS summary statistics for PUFAs were obtained from the UK Biobank, the data for DR were sourced from the Finnish database. Although all selected instrumental variables (IVs) were robust, potential bias could arise due to sample overlap [61]. Second, Our MR study demonstrated a causal relationship between genetically predicted PUFAs and DR; nevertheless, it is important to note that the findings of the MR analysis are based solely on genetic evidence. Third, we considered FAw3 or FAw6 components as a single entity and analyses of the causality of individual FAw3 or FAw6 members separately should be explored in future studies. These limitations highlight the need for continued research to address the potential biases arising from sample overlap to further examine the causal relationship between PUFAs and DR through complementary experimental approaches; additionally, the analysis of individual components of FAw3 and FAw6 has the potential to provide a more comprehensive understanding of the specific roles of these fatty acids in DR.

## 5. Conclusions

Our findings present initial genetic evidence to support a causal association between PUFAs and a reduction in DR risk. Specifically, the presence of both FAw3 and FAw6 demonstrates favorable effects on PDR, whereas FAw6 fatty acids play a mitigating role in BDR. However, neither type exhibits a causal relationship with SNPDR. Nonetheless, establishing an effective therapeutic approach for the administration of FAw3 and FAw6 in the context of DR remains an unanswered and critical question. Consequently, our comprehensive MR analysis strongly encourages further research to ascertain the precise roles and long-term effects of PUFAs, FAw3, and FAw6 on DR treatment.

## Figures and Tables

**Figure 1 nutrients-15-04208-f001:**
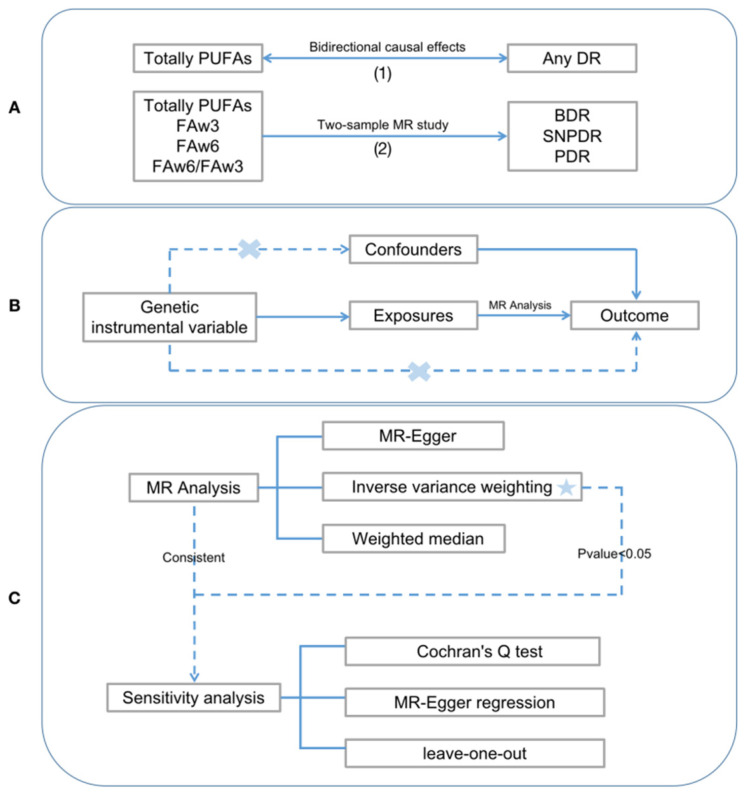
A study flame chart of the MR study revealing the causal relationship of PUFAs and DR. (**A**) The research direction in this study; (**B**) The principle of MR; (**C**) The process of MR Analysis. MR, Mendelian randomization; PUFAs, polyunsaturated fatty acids; FAw3, omega-3 fatty acids; FAw6, omega-6 fatty acids; DR, diabetic retinopathy; BDR, background diabetic retinopathy; SNPDR, severe non-proliferative diabetic retinopathy; PDR, proliferative diabetic retinopathy. Error symbol indicate no correlation; Asterisk indicates the most important.

**Figure 2 nutrients-15-04208-f002:**
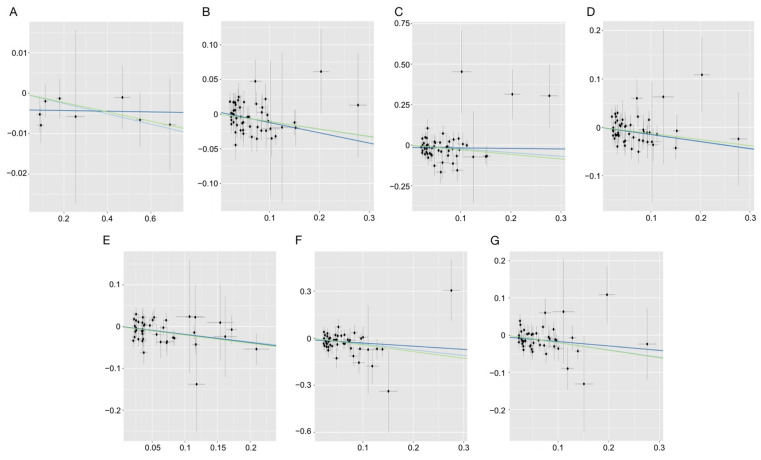
Scatter plots for MR analyses of the causal effects of PUFAs on DR. (Light blue represents IVW, dark blue represents MR-Egger regression, and green represents WM methods.) (**A**): ADR on total PUFAs; (**B**): total PUFAs on ADR; (**C**): total PUFAs on BDR; (**D**): total PUFAs on PDR; (**E**): FAw3 on PDR; (**F**): FAw6 on BDR; (**G**): FAw6 on PDR. MR, Mendelian randomization; PUFAs, polyunsaturated fatty acids; DR, diabetic retinopathy; ADR, any diabetic retinopathy; BDR, background diabetic retinopathy; PDR, proliferative diabetic retinopathy; FAw3, omega-3 fatty acids; FAw6, omega-6 fatty acids.

**Figure 3 nutrients-15-04208-f003:**
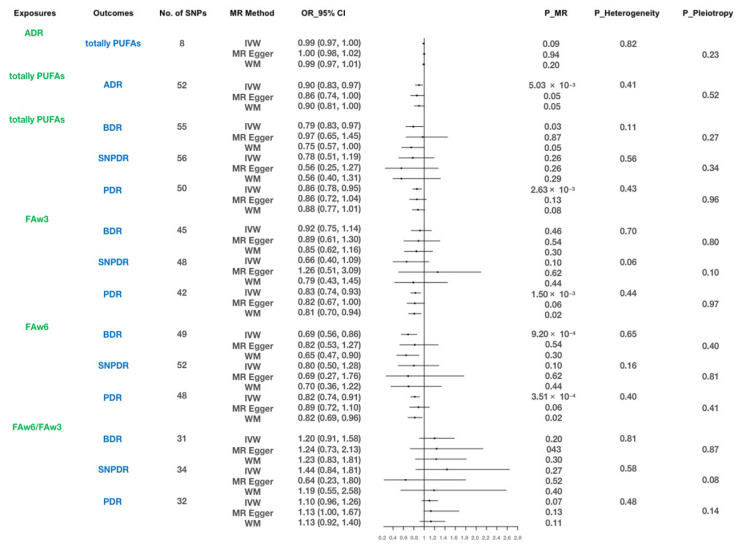
Study of association of PUFAs with DR using MR. SNP, single-nucleotide polymorphism; MR, Mendelian randomization; OR, odds ratio; CI, confidence interval; P_Heterogeneity, *p*-value for heterogeneity using Cochran Q test; P_Pleiotropy, *p*-value for MR-Egger intercept; PUFAs, polyunsaturated fatty acids; FAw3, omega-3 fatty acids; FAw6, omega-6 fatty acids; DR, diabetic retinopathy; BDR, background diabetic retinopathy; SNPDR, severe non-proliferative diabetic retinopathy; PDR, proliferative diabetic retinopathy.

## Data Availability

The datasets analyzed during the current study are available in the https://gwas.mrcieu.ac.uk and https://msk.hugeamp.org, accessed on 1 May 2023.

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
