# Peer review of "Mendelian Randomization Analysis Reveals Causal Effects of Polyunsaturated Fatty Acids on Subtypes of Diabetic Retinopathy Risk"

_nutrients, 2023, doi:10.3390/nu15194208_

Round 1

Reviewer 1 Report

Brief Overview:  Ren S. et al. conducted a comprehensive genetic analysis to determine the genomic association between all forms of diabetic retinopathy (DR) and efficacy of polyunsaturated fatty acids (PUFAs).  The authors performed bi-directional and two-sample Mendelian randomization utilizing very large sample sizes.  The authors found that high levels of PUFAs were associated with a reduced risk of any DR and proliferative DR (PDR). Further, the authors found that omega-3 PUFAs was mostly associated with a reduced risk of PDR while omega-6 PUFAs were associated with lower risks of background DR (BDR) and PDR.  The authors concluded that they provide genetic evidence for a causal relationship between PUFAs and a reduced risk for DR.

 Overall Comments:  This is a very interesting study and it addresses an important problem. Clearly the authors show that both omega -3 and -6 are important for maintaining ocular health and reducing the risk of DR.  Overall, the manuscript is well-written, and the methods and results are sound, particularly due to the large sample sizes which increases external and internal validity of the study.  The authors listed the limitations of the study and the conclusions are supported by the results.  This manuscript adds important information to the current literature regarding PUFAs and DR.

The following minor change is recommended:

State the working hypothesis in the Abstract and last paragraph of the Introduction.

Author Response

Dear Reviewer,

We sincerely appreciate your insightful and constructive review of our manuscript, ID number “nutrients-2605526”, originally titled “Causal Associations Between Polyunsaturated Fatty Acids and Diabetic Retinopathy: A Two-Sample Mendelian Randomization Study”, now changed to “Mendelian Randomization Analysis Reveals Causal Effects of Polyunsaturated Fatty Acids on Subtypes of Diabetic Retinopathy Risk.” 

Your positive feedback and recognition of the importance of our study in addressing diabetic retinopathy are greatly valued. Responses are provided below in bule to your comments.

Editorial Corrections:
• State the working hypothesis in the Abstract and last paragraph of the Introduction.
We have stated the working hypothesis in the Abstract (page 2, line 15 to 17) and the final paragraph of the Introduction (page 2, line 72 to 73), as suggested, to ensure clarity of our study's objectives. 

Additionally, we have expanded and optimized the manuscript based on comments from other reviewers. A manuscript with highlighted corrections labeled “nutrients-2605526-for Review Only” is attached to the revision of the manuscript and uploaded.

Once again, we sincerely appreciate your valuable feedback, which has undoubtedly enhanced the quality and clarity of our manuscript. Your insights are invaluable to us, and we are committed to addressing your suggestions to improve the overall presentation of our research.

Thank you for your time and consideration.

Sincerely,

Xiaorong Li
Tianjin Key Laboratory of Retinal Functions and Diseases, 
Tianjin Branch of National Clinical Research Center for Ocular Disease, Eye Institute and School of Optometry, 
Tianjin Medical University Eye Hospital, Tianjin, China
Email: [email protected]

Reviewer 2 Report

The MS by Ren and others shows a well designed study that provides new insight in a possible relationship between PUFAs and different types of DR (at least their phenotypes). It s worth publishing as an initial hypothesis with a very long run until practical results could be obtained.

N/A

Author Response

Dear Reviewer,

We sincerely thank you for your thoughtful review of our manuscript, ID number “nutrients-2605526”, originally titled “Causal Associations Between Polyunsaturated Fatty Acids and Diabetic Retinopathy: A Two-Sample Mendelian Randomization Study”, now changed to “Mendelian Randomization Analysis Reveals Causal Effects of Polyunsaturated Fatty Acids on Subtypes of Diabetic Retinopathy Risk.” 

We appreciate your positive feedback and recognition of the study's design and potential contributions. Responses are provided below in bule to your comments.

Comments:
• Minor editing of English language required.
We have completely revised the original manuscript and obtained the corresponding certificates. However, since only one word file is allowed to be uploaded, we did not attach the certificate.

• The methods can be improved.
We extend the content and optimize the structure of the Method to make it clearer and more logical on page 3 to 6.

Additionally, we have expanded and optimized the manuscript based on comments from other reviewers. A manuscript with highlighted corrections labeled “nutrients-2605526-for Review Only” is attached to the revision of the manuscript and uploaded.

We once again express our sincere appreciation for your time and consideration. Your feedback is invaluable in propelling our research, and we eagerly anticipate contributing further insights to this field in the future.

Thank you for your support.
Sincerely,

Xiaorong Li
Tianjin Key Laboratory of Retinal Functions and Diseases, 
Tianjin Branch of National Clinical Research Center for Ocular Disease, Eye Institute and School of Optometry, 
Tianjin Medical University Eye Hospital, Tianjin, China
Email: [email protected]

Reviewer 3 Report

Ren et al conducted a bidirectional two-sample Mendelian randomization investigation to delve into the intricate dynamics of exposure-outcome relationships. Authors utilized assessments for pleiotropy, heterogeneity, and sensitivity. They found elevated levels of total polyunsaturated fatty acids were linked to a diminished risk of both ADR and PDR. Furthermore, FAw3 demonstrated a significantly reduced risk of PDR, whereas FAw6 showcased associations with lowered risks for both BDR and PDR. This study is interesting as these findings present compelling genetic evidence, marking a watershed moment as they establish, for the first time, a causal relationship between PUFAs and the reduced risk of diabetic retinopathy.

Manuscript is interesting but I would encourage to rewrite the manuscript as most of the parts overlap with the previous studies (https://www.sciencedirect.com/science/article/pii/S0002916522105344?via%3Dihub#fig1).

I would recommend this work only after "rewriting" revision(Including change in the title).

Manuscript is interesting but I would encourage to rewrite the manuscript as most of the parts overlap with the previous studies (https://www.sciencedirect.com/science/article/pii/S0002916522105344?via%3Dihub#fig1).

I can recommend this work only after writing revision(Including change of the title).

Author Response

Dear Reviewer,

We sincerely appreciate your thoughtful review of our manuscript, ID number “nutrients-2605526”, originally titled “Causal Associations Between Polyunsaturated Fatty Acids and Diabetic Retinopathy: A Two-Sample Mendelian Randomization Study”, now changed to “Mendelian Randomization Analysis Reveals Causal Effects of Polyunsaturated Fatty Acids on Subtypes of Diabetic Retinopathy Risk.” 

Your feedback and comments are invaluable, and we have incorporated your suggestions into our revisions. Responses are provided below in bule to your comments.

Comments:
• This study is interesting as these findings present compelling genetic evidence, marking a watershed moment as they establish, for the first time, a causal relationship between PUFAs and the reduced risk of diabetic retinopathy.
Thank you for your confirmation very much.

• Extensive editing of English language required.
We have completely revised the original manuscript and obtained the corresponding certificates. However, since only one word file is allowed to be uploaded, we did not attach the certificate.

• Does the introduction provide sufficient background and include all relevant references? -Must be improved.
We have expanded and optimized the introduction on pages 2 (line 42 to 57) to make it more substantial and to emphasize its novelty and distinctiveness, and ensured a fresh perspective on the topic.

• Manuscript is interesting but I would encourage to rewrite the manuscript as most of the parts overlap with the previous studies (https://www.sciencedirect.com/science/article/pii/S0002916522105344?via%3Dihub#fig1).
I can recommend this work only after writing revision(Including change of the title).
1)We have appropriately cited the well-written article on page 2 (line 71, reference 23) you mentioned, which overlaps with our manuscript. 
2)Revisions have been made to our Introduction to address gaps and improve structure. 
3)We extend the content and optimize the structure of the Method to make it clearer and more logical on page 3 to 6.
4)The manuscript's title has been revised to “Mendelian Randomization Analysis Reveals Causal Effects of Polyunsaturated Fatty Acids on Subtypes of Diabetic Retinopathy Risk.”
5)Although there may be some similarities in content due to the shared use of Mendelian randomization methods and a focus on polyunsaturated fatty acids as exposures, our outcomes are significantly different. Their study pertains to kidney-related content, while ours focuses on different subtypes of diabetic retinopathy. We have made every effort to adjust and modify where necessary, and we kindly request your understanding and guidance for any remaining areas that may need further refinement.

Additionally, we have expanded and optimized the manuscript based on comments from other reviewers. A manuscript with highlighted corrections labeled “nutrients-2605526-for Review Only” is attached to the revision of the manuscript and uploaded.

Once again, we sincerely appreciate your valuable feedback, which has undoubtedly improved the quality and clarity of our manuscript. Your insights are invaluable, and we remain committed to addressing your suggestions to enhance the overall presentation of our research.

Sincerely,

Xiaorong Li
Tianjin Key Laboratory of Retinal Functions and Diseases, 
Tianjin Branch of National Clinical Research Center for Ocular Disease, Eye Institute and School of Optometry, 
Tianjin Medical University Eye Hospital, Tianjin, China
Email: [email protected]

Reviewer 4 Report

The well-structured research article contains appropriate methods, tables, and detailed figures illustrating the results and limitations of the study. 

Author Response

Dear Reviewer,

We sincerely appreciate your thoughtful review of our research article, ID number “nutrients-2605526”, originally titled “Causal Associations Between Polyunsaturated Fatty Acids and Diabetic Retinopathy: A Two-Sample Mendelian Randomization Study”, now changed to “Mendelian Randomization Analysis Reveals Causal Effects of Polyunsaturated Fatty Acids on Subtypes of Diabetic Retinopathy Risk.” 

Your positive feedback on the manuscript's structure, content, and the use of appropriate methods, tables, and figures is greatly valued. 

Additionally, we have expanded and optimized the manuscript based on comments from other reviewers. A manuscript with highlighted corrections labeled “nutrients-2605526-for Review Only” is attached to the revision of the manuscript and uploaded.

Thank you for your time and consideration. Your insights are invaluable, and we eagerly anticipate making further contributions to the field of polyunsaturated fatty acids and diabetic retinopathy.

Sincerely,

Xiaorong Li
Tianjin Key Laboratory of Retinal Functions and Diseases, 
Tianjin Branch of National Clinical Research Center for Ocular Disease, Eye Institute and School of Optometry, 
Tianjin Medical University Eye Hospital, Tianjin, China
Email: [email protected]
